# Dog Ecology, Bite Incidence, and Disease Awareness: A Cross-Sectional Survey among a Rabies-Affected Community in the Democratic Republic of the Congo

**DOI:** 10.3390/vaccines7030098

**Published:** 2019-08-26

**Authors:** Céline Mbilo, Jean-Baptiste Kabongo, Pati Patient Pyana, Léon Nlonda, Raymond Williams Nzita, Bobo Luntadila, Badivé Badibanga, Jan Hattendorf, Jakob Zinsstag

**Affiliations:** 1Swiss Tropical and Public Health Institute, P.O. Box, CH-4002 Basel, Switzerland; 2University of Basel, Petersplatz 1, CH-4001 Basel, Switzerland; 3Université Pédagogique Nationale de Kinshasa, BP 8815 Kinshasa, Congo; 4Institut National de Recherche Biomédicale (INRB), Avenue de la Démocratie, BP 1197 Kinshasa/Gombe, Congo; 5Clinique Vétérinaire d’Etat de Matadi, Matadi, Congo

**Keywords:** Rabies, free-roaming dog, dog ecology, dog bite incidence, zoonoses, Democratic Republic of the Congo

## Abstract

Despite the existence of safe and efficacious human and animal rabies vaccines, millions of people remain at risk of exposure to this deadly zoonotic disease through bites of infected dogs. Sub-Saharan African countries, such as the Democratic Republic of the Congo (DRC), bear the highest per capita death rates from rabies where dog vaccination and availability of lifesaving post-exposure prophylaxis (PEP) is scarce. Mass dog vaccination is the most cost-effective and sustainable approach to prevent human rabies deaths. We conducted a cross-sectional household survey in a rabies-affected community in Matadi, DRC, to estimate the size of the owned dog population and dog bite incidence and assess knowledge and practices regarding rabies, as preparation for future mass dog vaccination campaigns. Our study revealed that the owned dog population in Matadi was almost ten times larger than assumed by local veterinary officials, with a large proportion of free-roaming unvaccinated dogs. The annual dog bite incidence of 5.2 per 1000 person years was high, whereas community rabies knowledge was low resulting in poor practices. Given these findings, human rabies deaths are likely to occur in this community. Lack of disease awareness could negatively affect participation in future mass dog vaccination campaigns. A public sensitization campaign is needed to promote appropriate rabies prevention (washing bite wounds and PEP) and control (dog vaccination) measures in this community.

## 1. Introduction

Rabies, a fatal viral disease transmitted through the bite or scratch of an infected animal, is estimated to cause about 21,500 human deaths per year in Africa [1]. Domestic dogs are the main reservoir species of the rabies virus (RABV) and by far the most common transmitter of the disease to humans [1]. Because most human rabies exposures (>99%) result from dog bites and highly efficacious dog rabies vaccines are available, the disease in humans can be prevented through vaccination of its animal source.

In 2018, the Global Strategic Plan to end human deaths from dog-mediated rabies by 2030 was launched by the United Against Rabies (UAR) collaboration of four international organizations, the World Health Organization (WHO), the World Organisation for Animal Health (OIE), the Food and Agriculture Organization of the United Nations (FAO), and the Global Alliance for Rabies Control (GARC) [2]. The global strategic plan emphasizes mass dog vaccination as key for sustainably preventing human rabies, along with increasing disease awareness and ensuring prompt post-exposure prophylactic treatment (PEP) for bite victims.

Rabies transmission among dogs (and therefore exposure to humans) can be interrupted when a sufficient proportion of the dog population is immunized [3]. The threshold proportion p_c_ of the population that needs to be immunized to interrupt transmission under assumptions of homogeneity is determined by the basic reproductive number R_0_ of an infectious disease as p_c_ = 1 − 1/R_0_. R_0_ of rabies transmission between dogs is estimated to be relatively low, between 1.05 and 1.72 [4,5]. Based on empirical data and disease modeling, the UAR collaboration recommends annual vaccination campaigns that attain at least 70% of the dog population to sustain interrupted transmission [2,3,6]. Recent work based on social contact networks of dogs confirmed the recommended coverage of 70% [7]. Therefore, the size of the target dog population is a critical factor to consider in planning vaccination campaigns. A straightforward approach to obtain a rough estimate of the owned dog population is a household survey to determine the dog/human ratio, with subsequent extrapolation to a given area [8].

Even though the dog rabies vaccine is highly efficacious, final effectiveness in the field, in terms of vaccination coverage, depends on accessibility of the dog population for vaccination and many other effectiveness factors [9,10,11,12,13], which are influenced by the sociocultural context [14]. Accessibility of dogs for parenteral vaccination is affected by their ownership status. Increasing evidence suggests that the majority of dogs in Africa are owned but allowed to roam freely and that the proportion of ownerless dogs is low [8,15]. Proof-of-concept studies in Africa were successful in reducing human and animal rabies cases but mainly targeted the owned dog population [14,16,17,18,19]. Lack of disease awareness and inability of dog owners to handle their dog can be a constraint to participation in vaccination campaigns [10,14]. 

While dog vaccination is essential to sustainably control rabies in the reservoir population and prevent human deaths, individuals also need to be aware of preventive measures following an animal bite, such as seeking PEP and informing the veterinary service of suspected rabid animals. It is thus critical to understand the local context and assess community awareness on rabies prior to implementation of mass dog vaccination campaigns. Knowledge, attitude, and practices (KAP) studies within rabies-affected communities expose factors that influence response to vaccination campaigns and identify knowledge gaps and practices that hamper rabies prevention and control.

Dog bites have been widely used as a proxy to estimate human rabies deaths based on decision-tree probability models [1,20,21,22]. Epidemiological data on dog bites are often obtained from health facilities, but not all dog bite victims seek care in the formal health care sector. Therefore, such data fail to capture the true incidence of dog bites in the community and are likely to be biased towards more severe bites that require medical attention [23]. Community population-level estimates of actual dog bite incidence are needed to assess the burden of rabies in a given area.

In the Democratic Republic of the Congo (DRC), first reports of rabies date back to 1923 [24] and recurrent outbreaks have occurred across the country ever since [25,26,27]. In 2009, 21 clinically-confirmed rabies cases in children were registered in the Pediatrics Department of a single hospital in the capital city of Kinshasa in only seven months. In a retrospective study of 5053 animal attacks in Kinshasa between 2009 and 2013, 2.5% were likely due to rabid animals [28]. Despite this alarming situation, no official rabies control program is in place and very limited data on dog population and community awareness regarding rabies exist. In the DRC, dog vaccination is not governmentally subsidized. The Congolese National Order of Veterinarians (ONMVC: Ordre National des Médecins Vétérinaires du Congo) fixed the price for dog rabies vaccination at no less than 20 USD [29], although it has been shown earlier in a similar setting that cost above 2 USD cannot be borne by the community [30]. Recent studies found dog vaccination coverage between 24% and 81% in different communes of Kinshasa [29,31]. The vaccination status of dogs appeared to depend on access to vaccine and the dog owner’s economic situation [29,31]. About two-thirds of dogs were allowed to roam freely, and the proportion of ownerless dogs was estimated to be less than 2% [31]. 

In this article, we report the findings from a cross-sectional household survey conducted in a rabies-affected community in the DRC to study both the human and dog population as a preparation for future dog mass vaccination campaigns in the DRC. The aims of the study were to collect baseline data on the size and characteristics of the owned urban dog population, assess community knowledge and practices regarding rabies and, provide a community-level estimate of dog bite incidence. 

## 2. Methods

### 2.1. Study Site

Our study took place in Matadi (5°49′03″S 13°28′15″E, Figure 1), the capital of the province of Kongo Central, located in the western part of the DRC. Matadi was selected due to the occurrence of several rabies outbreaks over the past few years (personal communication, government veterinary clinic of Matadi) and its accessibility by road from Kinshasa. Compared to other provinces, Kongo Central is relatively well developed as it is the only province with access to the Atlantic Ocean and harbors DRC’s two main ports. In the second Demographic and Health Survey (DHS), 81% of the households in Kongo Central were categorized into the three highest quintiles of the wealth index [32]. This is still low in comparison to Kinshasa, where over 97% of households are classified in the highest quintile of the wealth index [32]. Matadi is located on the Congo River and divided into three communes comprising 17 neighborhoods. According to the 2015 annual report of the municipality, an estimated 301,644 people live in Matadi across 110km^2^. The government veterinary clinic of Matadi recorded 873 dogs living in the city in 2017 (personal communication, 2018).

### 2.2. Data Collection

In May 2017 (dry season), we conducted a cross-sectional household survey among the residents of Matadi. A structured questionnaire in French based on the Report of WHO Consultation on Dog Ecology Studies Related to Rabies Control [33] and the Guidelines for dog population management [34] was developed and pre-tested in 40 Matadi households during an exploratory visit in July 2016. The questionnaire consisted of three sections. The first part contained questions on household sociodemographic characteristics, while the second part gathered information on the respondent’s knowledge and practices regarding rabies and animal bite incidents occurring in the household in the last three years. The third part captured detailed data on individual dogs older than three months owned by the household. A household was defined as one person living alone or a group of persons, related or not, occupying a housing unit (house, apartment, a single or group of rooms), with shared living accommodation and meals under the lead of a household head.

### 2.3. Sample Size Calculation and Household Sampling Procedure

Due to the lack of a household register or clearly defined administrative boundaries of neighborhoods, we applied the sampling procedure proposed by Schelling and Hattendorf [35] using random geo-coordinates. A set of numbered random geo-coordinates that served as interview starting points for the survey teams was generated with R software version 3.3.1 [36] and exported as a kml file to Google Earth software version 7.1.2.2041. Points lying outside the city boundary were discarded.

Based on expertise gathered from research projects in Bamako (Mali), Iringa (Tanzania) and N’Djamena (Chad) [37,38,39], we assumed a prevalence of dog owning households of 14%. To obtain a point estimate with a precision, defined as one half-length of the confidence interval, of 4.5 percentage points, we needed to survey 50 clusters with 20 households each, assuming an intra-cluster correlation coefficient of 0.2. 

Four field teams conducted the house-to-house survey. Each team consisted of an interviewer and a local guide. Three interviewers were recruited at the veterinary faculty of the National Pedagogical University in Kinshasa. One interviewer and four veterinary technicians who served as local guides were recruited at the government veterinary clinic of Matadi. Teams were trained on the household selection process and administration of the questionnaire during a pre-test in July 2016 and two days prior to the launch of the survey in May 2017. The head of the government veterinary clinic of Matadi supervised and coordinated the four field teams. After pre-testing, one question regarding household sanitary situation was removed from the questionnaire because it was considered inappropriate by respondents. Several questions were rephrased to improve understanding. Based on the experience gained during the pre-test, we deemed data collection in 21 households per team and day attainable and estimated study completion within 13 days. 

The study lasted from May 15 to May 28, 2017. On the survey days, the four field teams gathered at the government veterinary clinic of Matadi at 07:30 AM. Each team was guided to a random geo-coordinate starting point using the free, open source mobile application MAPS.ME. Facing north, the nearest household on the right side of the street was chosen for a first interview, then every third household was interrogated. At cross roads, the team first turned left and then right at the next cross road and so on. After completion of seven households, the teams moved back to the starting point and repeated the same procedure facing east, south, and west until a total of 21 households were visited. If for any reason the survey could not be carried out in a household (e.g., nobody is at home, the person refuses to participate or was too young, the house was a shop), the neighboring household continuing in the direction of movement was selected. The questionnaire was administered to the household head or another occupant over 16 years of age. Depending on the respondent’s preference, interviews were conducted in French or the local languages Lingala or Kikongo. Each interviewer was equipped with a computer tablet (Trekstor SurfTab breeze) configured with Open Data Kit (ODK) software and a solar battery charger. In non-dog owning households, interviews were limited to the first two parts of the questionnaire. At the end of each survey day, completed questionnaires were downloaded from the tablets to a password protected external hard drive. 

### 2.4. Data Analysis

Data collected during the study were downloaded from ODK software in comma-separated value files and analyzed using R software version 3.5.0 [36]. Maps were created with QGIS version 3.4.4. 

We applied generalized estimating equations (GEE) for binary distributed outcome variables, i.e., dog ownership status (yes/no), having heard of rabies (yes/no), rabies knowledge (adequate/inadequate), and history of animal bite in the household (yes/no), with a logit link function and independent correlation structure to account for the clustered nature (random geo-coordinates) of the questionnaire data. Association with explanatory variables (age, sex, education, household position, livestock ownership, dwelling ownership status, water source, residence) were analyzed using univariable and multivariable analysis. All explanatory variables with p-values <0.2 in univariable analysis were included in the multivariable model [40]. Estimates are presented as odds ratios (OR) and adjusted OR with corresponding 95% confidence interval (CI). The significance level was set at p-value≤0.05. To estimate the dog/human ratio, we used GEE for negative binomial distributed outcomes and independent correlation structure to account for the over-dispersed nature of the distribution of dog counts per household. The point estimate and CI were divided by the mean number of persons per household.

We constructed knowledge and practices scores for respondents who had heard of rabies (*n* = 551) based on eight and three questions, respectively (see Appendix A). All questions were equally weighted with a maximum score of 4 per knowledge and 3.5 per practices question. No penalty was given for incorrect answers. A respondent could obtain overall scores of 32 for rabies knowledge and 10.5 for practices, when all questions were correctly answered. Respondents that obtained 16 points or higher were classified as knowledgeable about rabies. We assessed the relationship between the practices score (outcome variable) and knowledge score (explanatory variable) using GEE for normal distributed outcome variables with an identity link function and independent correlation structure. 

Based on proxy markers similar to those used in the DHS conducted in the DRC in 2013–2014, we assessed the socioeconomic status (SES) of households [29,32]. We used the following five housing characteristics: availability of piped drinking water, wall material, roof type, livestock ownership, and fenced property. A score of 0 or 1 was attributed to each item: 0 in case of unavailability of piped water, straw/wooden wall, straw roof, non-ownership of livestock, and unfenced property, and 1 in case of availability of piped water, cement/brick wall, sheet roof, ownership of livestock, and fenced property. Households that scored ≤ 2 points were assigned low SES, while households that scored ≥ 3 were classified as middle SES. Our SES was well in line with the DHS wealth index [32].

### 2.5. Ethical Approval

We obtained research permission from *EKNZ* (Ethics Committee *for* Northwest/Central Switzerland, EKNZ BASEC Req-2017-00395) *and* ethical clearance from the Ministry of Public Health ethics committee at the Clinique Ngaliema, Kinshasa (COMETH/TKK/PRES 002/2017). Study participants were verbally informed about the purpose of the survey, and we obtained verbal consent from each participant before administering the questionnaire. Respondents were free to refuse or discontinue participation at any time. Before the beginning of the study, a meeting was held with the mayor of Matadi, who provided written permission to conduct the study.

## 3. Results

### 3.1. Household and Respondent Characteristics

In May 2017, we approached a total of 1095 households in the city of Matadi of which 1056 completed the survey (96%). The total number of occupants living in the 1056 households was 6742 with a median household size of 6 (IQR: 4–8). Details about household and respondent characteristics are given in Table 1. More women (63%) were interviewed and about half (44%) of the interviews were conducted with the wife of the head of household. Most respondents (90%) had secondary (78%) or higher education (12%) and around half (49%) were older than 40 years. Two-thirds of households were owned and slightly more than half (53%) were located in peri-urban parts of the city. Almost all households (97.5%) had no fence around their premises, meaning that dogs could enter and leave the premises without restriction. Two-thirds of households (67%) disposed of garbage so that it was easily accessible for roaming dogs. 

### 3.2. Dog Ownership and Dog-Keeping Practices

The percentage of dog-owning households (DOHH) was 9.5%, with 1–9 dogs per DOHH (median: 1, IQR: 1–2). We recorded a total of 178 dogs during the household survey. The dog/human ratio, when considering the 6742 persons represented in this survey, was 1:37.7 (95% CI: 32.1–44.3). An estimate of the owned dog population based on an extrapolation of the dog/human ratio predicts 8001 dogs for the city of Matadi (95% CI: 6809–9397), resulting in a dog density of 74/km^2^. 

More than two-thirds of dog owners (63%) allowed their dogs to roam freely at all times, while the remainder confined their dogs part time either during the day (35%) or night (2%). Dogs were primarily kept for security (92%) and livestock herding (7%). Only one household (1%) reported using dogs for hunting. The majority of dogs (93%) were fed with leftovers from human consumption and/or slaughterhouse (16%), while 3% of dogs were provided commercial dog food. Less than one in four dog owners (22%) stated that they provided some level of veterinary care to their dogs. Dog owners reported that their dogs would spend most of their time on (86%) or in front of the dog owner’s premises (8%), with only a small minority roaming the streets (6%). The most common way to acquire a dog was through purchase (52%) or as a gift (30%), followed by offspring from another household dog (15%) or adoption from the street (3%). Among households that purchased a dog, about two-thirds (58%) were bought in another neighborhood of the city, 36% were acquired in the same neighborhood and 6% were purchased outside the city. 

When asked if ownerless dogs were present in the neighborhood, the majority of respondents (82%) stated that there were no ownerless dogs, while 17% reported having seen ownerless dogs and 1% could not provide a response to this question. The median number of ownerless dogs stated by 154 out of 182 respondents who reported having seen ownerless dogs was 2 (IQR: 1–4), the remaining 28 respondents did not provide an estimate of the ownerless dog population in the neighborhood.

Among non-dog owning households (956/1056) reasons given for non-ownership were a dislike of dogs (39%), no necessity for keeping a dog (18%), not enough space (13%), or no specific reason (10%). Only 8% stated that they were afraid of the public health risks associated with dog-ownership. The remaining 12% provided various other reasons.

#### Factors Associated with Dog Ownership

The results of the univariable and multivariable analysis of factors associated with dog ownership are presented in Table 2. In the final multivariable model, livestock owning households had twice the odds of owning a dog as compared to households without livestock (95% CI: 1.25–3.23, *p* = 0.004). Similarly, the odds of owning a dog were greater among dwelling owners as compared to tenants (adj OR = 2.37, 95% CI: 1.36–4.15, *p* = 0.002). We found no association with residence or household socioeconomic status. 

### 3.3. Dog Population Characterisics

Of the 178 dogs identified during the household survey, 57 (32%) were puppies (0–3 months), 31 (17%) young dogs (4–11 months), 89 (50%) adult dogs (1–7 years), and 1 old dog (> 7 years). Detailed information on 106 dogs older than 3 months was available, while data on 3 young dogs and 12 adult dogs were missing. The study dog population consisted of 53 males (50%) and 53 females (50%) with a sex ratio of exactly 1:1 (Figure 2). The 57 puppies were not sexed. None of the female dogs and only 3 out of 53 male dogs were neutered. Roughly three-quarters of dogs (71%) were local nondescript breeds, followed by mixed breeds (29%). Twenty-six out of 53 (49%) females gave birth in the 12 months preceding the study with a median litter size of 6 puppies (IQR: 4–7). Of the 142 puppies born, 37% died, and the remainder were kept in the household (42%), sold (11%) or given away (10%). 

#### Dog Vaccination and Accessibility for Parenteral Vaccination

According to dog owners, 25 out of 106 dogs (24%) were vaccinated against rabies at some point in their lifetime. Of these, only three-quarters of dogs (72%, 18/25) had a vaccination booklet and one-third of vaccinations (33%, 6/18) were outdated at the time of the survey. Vaccination status of the 57 puppies was not available. When asked about reasons for non-vaccination, almost one-third of dog owners (30%) refused to respond to this question. One in three dog owners lacked the money to purchase the vaccine, while 16% did not know where to find the vaccine and 14% thought that their dog was too young for vaccination. About 7% of dog owners did not know that their dog should be vaccinated against rabies. Dogs living in households assigned a high SES were more often vaccinated than dogs owned by low SES households (30% vs. 0%). Three-quarters of dog owners (74%) claimed that they would be willing to spend money on dog vaccination and the median amount they would be willing to pay for dog vaccine was 2.1 USD (IQR: 0-3.6 USD). The majority of dog owners (92.5%) stated that they could handle their dog for parenteral vaccination. 

### 3.4. Animal Bite Incidents

Almost one in ten households (9%) had at least one family member who experienced an animal bite in the previous three years. A total of 97 bite victims were reported, of which 62 % were adults and 38% children. Sex of the bite victim was not recorded. We estimated a dog bite incidence of 5.2 per 1000 person years (95% CI: 4.3–6.2). All bite victims were bitten by dogs, which were in most cases (89%) known by the bite victim, reported as a neighbor’s dog (60%), a dog owned by the household (16%) or a community dog (13%). Only 7% were bitten by an unfamiliar ownerless dog. Dog ownership was positively associated with the occurrence of a bite incident in the household (OR = 2, 95% CI: 1.01–3.85, *p* = 0.034). More than one in ten bite victims (11%) died before the survey, and 64% of respondents stated that the death was related to the bite incident, i.e., the bite victims died after showing symptoms of rabies. 

### 3.5. Community Rabies Knowledge

When asked if respondents knew of diseases transmitted by dogs, the most common (37%) disease mentioned was rabies, followed by tetanus (14%). Slightly more than half of respondents (52%) had previously heard of rabies, with the majority (95%) correctly describing rabies as a disease. Only study respondents who had heard of rabies were further questioned (*n* = 551). Most respondents (88%) knew that rabies was transmitted through the bite of an infected animal but were unaware of other modes of transmission. Almost all respondents (94%) mentioned dogs as the principal reservoir of rabies, followed by cats (19%). Forty percent of respondents correctly described the symptoms of rabies (mentioned one or more symptoms) in animals. Aggressiveness (95%) was the most commonly mentioned symptom, followed by behavior change (40%). Figure 3 gives an overview of animal rabies symptoms stated by the participants. About half of respondents (45%) were able to state one or more rabies symptoms in humans. A wide range of symptoms were identified by the study participants, with monologizing (23%), insanity (23%), barking like a dog (22%) and aggression (8%) most frequently mentioned. Of the 551 respondents, 33% claimed to know someone who died of rabies and 30% stated that they had encountered a rabid animal. The odds of correctly describing animal rabies symptoms were 3.43 times higher among respondents who had seen a rabid animal (95% CI: 2.14–5.5, *p* < 0.001). There was no similar association found between knowing someone who died of rabies and correctly describing human rabies. Around three-quarters of respondents (73%) stated that rabies is a preventable disease, and half of respondents (51%) knew that human rabies is preventable through dog vaccination. However, only 3% of respondents mentioned human vaccination as a preventive measure and only 14% knew that rabies is fatal following the onset of symptoms.

#### 3.5.1. Factors Associated with Rabies Knowledge

The mean knowledge score was 13.65 (SD: 4.85, median: 13), with a minimum of 0 and a maximum of 26 points. The most frequent knowledge scores were 12 and 13 (11.4% and 17.4%, respectively). Using the cut-off score of ≥ 16 to classify high and low level of rabies knowledge, 189 out of 551 (34%) respondents were classified as having a high knowledge about rabies. Results of the multivariable analysis indicated that rabies knowledge was higher among respondents who were i) male, ii) 40 years or older, and iii) had a high level of education, whereas urban residence was negatively associated with knowledge (Table 3). 

#### 3.5.2. Practices towards Suspected Rabid Animals

When asked about measures they would take when encountering a rabid animal, two-thirds (62%) of respondents would kill the animal, while 15% would chase it away or run from it and 3% would do nothing. Twenty percent of respondents would inform the veterinary service of the suspected rabid animal. Only 2% of respondents would take the carcass of a suspected rabid to the veterinary service for laboratory testing, while the majority would bury it (82%) or throw it away (16%). 

#### 3.5.3. Health Seeking Behavior

After an animal bite, only 2% of respondents would wash the wound with water as a first aid measure. Respectively, 85% and 4% would report to a health facility or the veterinary service immediately after a bite. Only 2% would seek anti-rabies vaccine, while 6% would get anti-tetanus vaccination. A small fraction would contact a traditional healer (0.5%) or do nothing after a bite (0.5%). When asked about where to find human rabies vaccine, respondents stated at a health facility (41%) and/or veterinary service (39%). One-fifth of respondents indicated that they did not know where to obtain human rabies vaccine.

#### 3.5.4. Relationship between Rabies Knowledge and Practices

Overall practices scores were low with a mean of 3.11 (SD: 1.05, median: 3.5), minimum of 0, and maximum of 6.5 out of 10.5 possible points. We found a significant positive correlation between the knowledge and practices score (Beta = 0.056, *p* < 0.001). Figure 4 shows the relationship between the knowledge and practices score by level of education. 

## 4. Discussion

Despite recurrent rabies outbreaks across the DRC, the disease remains neglected with few rabies-focused studies limited to Kinshasa [26,28,29,31]. Controlling rabies in the animal reservoir through mass dog vaccination is the most cost-effective and sustainable approach to prevent dog-mediated human rabies cases [2,42,43]. Knowledge of the dog population size is a prerequisite for adequately planned vaccination campaigns, and understanding community rabies awareness is critical to adapt rabies interventions to the local context and ensure high community acceptance and participation. This study revealed several important findings that help plan prospective dog rabies awareness raising and vaccination campaigns in the city of Matadi and DRC urban areas. First, the owned dog population in Matadi was almost ten times larger than assumed by veterinary officials. Secondly, a large proportion of the owned dog population was unvaccinated and free-roaming. Thirdly, the annual community-level dog bite incidence was high as compared to other sub-Saharan African settings. Lastly, rabies awareness among the community was low with knowledge gaps that result in poor practices. 

### 4.1. Dog Ownership and Dog-Keeping Practices

In our study, about one in 10 households owned at least one dog. This proportion of DOHH is comparable to studies carried out in neighboring countries. For example, 7% and 14–15% of households in coastal and inland urban areas in Tanzania [38,44] and 11% of households in a suburb of the capital city of Zambia owned dogs [45]. In other sub-Saharan African areas, the proportion of urban DOHH was higher (17% in South Africa [46], 28% in Chad [37], 61% in Ethiopia [47], 89% in Madagascar [48], and 95% in Nigeria [49]). An exception in West Africa was Mali, where only 9% of households in the capital city of Bamako owned a dog [39]. The observed dog/human ratio of 1:37.7 is lower than the dog/human ratio of 1 to 21 (95% CI: 12.5–37.1) determined by a meta-analysis for urban Africa [50] and generally lower than dog/human ratios found in other urban sub-Saharan African areas (Chad: 1:20.7 [37], Tanzania: 1:14 [38,44]. South Africa: 1:12.7 [46], Nigeria: 1:3.5-5.7 [49,51], Madagascar: 1:4.5 [48]) with two exceptions, Zambia (1:45) [45] and Mali (1:121) [39]. In rural areas, dog/human ratios are generally higher but the dog density per km^2^ is lower [8]. Based on the dog/human ratio, we estimated the owned dog population of Matadi at 8113 individuals. This was almost ten times higher than the official number of 873 assumed by the government veterinary clinic of Matadi. This finding has large financial implications for planning prospective dog vaccination campaigns, in terms of vaccine doses and human resources needed to reach the 70% vaccination coverage of recommended by the UAR collaboration. Although our data indicates a tenfold underestimation of the owned dog population, this figure is still a rough estimate of the actual dog population size and should be refined after vaccination campaigns using post-vaccination transects as suggested by Sambo, et al. [52]. The study compared different methods to estimate dog populations and found that estimates based on household surveys were often imprecise due to relatively small sample sizes. 

One limitation of our study is that the method used to estimate the dog population size did not take into account the unowned dog population. The majority of study participants (82%), however, reported that there were no ownerless dogs in their neighborhood, which could indicate that the proportion of ownerless dogs is low. This result is further corroborated by a study conducted in Kinshasa which found that ownerless dogs accounted for less than 2% of the entire dog population [31] and the widely accepted assumption that the proportion of unowned dogs across sub-Saharan Africa is low [8,38].

Many of our findings concerning dog keeping practices agree with previous studies conducted in urban sub-Saharan Africa, specifically that the majority of dogs were free-roaming [37,38,39,45,46,47,48,51], fed regularly by their owners [47,48,49,53,54], and primarily kept for security and livestock herding [37,44,45,47,48,49,53,54]. In contrast, in a municipality of Kinshasa, more than half of the surveyed dogs were confined [29]. A possible explanation is the relative higher standard of living in Kinshasa compared to Matadi. Dogs in middle- and high-income residential areas in Kinshasa are typically kept within fenced properties to guard the premises [29]. Only a fraction (2.5%) of the properties in Matadi had fences that kept dogs from roaming. This difference in SES between the capital city and Matadi is also reflected in the preference of residents of Kinshasa for hybrid breeds (purchase price about 400 USD per puppy) [29], whereas the majority of dogs in Matadi were local nondescript breeds. 

### 4.2. Dog Vaccination and Accessibility

Although most dogs could roam freely, they remained mainly on (86%) or in front (8%) of the dog owner’s premises, indicating that most dogs would be accessible during a door-to-door vaccination campaign. Almost all dog owners (92.5%) were confident they could handle their dog for parenteral vaccination. During a vaccination campaign conducted in Matadi on World Rabies Day 2017, no problems restraining dogs and no dog bites were noted [29]. However, difficult dogs might not have been presented for vaccination. Twenty-five out of 106 (24%) dogs were reported to be vaccinated against rabies but only 11% had a valid vaccination booklet. This is far below the 70% coverage recommended by the UAR collaboration and the reported vaccination coverages from Kinshasa, which varied between 24%–81% [29,31]. Kazadi et al. [29] explained the high vaccination coverages in Kinshasa as due to increased willingness of dog owners in middle- and high-income neighborhoods to pay for vaccination of expensive hybrid dogs. In Matadi, dog owner economic status seems to partially affect dog vaccination status. One in three dog owners stated that they lacked the money for vaccination and dogs in households with a middle SES were more often reported as vaccinated compared to dogs from low SES households (30% vs. 0%). Almost one-third of respondents refused to answer why their dog was not vaccinated, which could be explained as fearing legal consequences for non-vaccination. On average, dog owners would be willing to pay 2.1 USD for dog vaccination, which is in line with findings from Chad [55] and ten times less than the vaccination price of US$ 20 fixed by the Congolese National Order of Veterinarians. Although per dog vaccination cost seems to vary between different regions [11], the mean price willing to pay is below the actual per dog vaccination cost of $ 4.9–5.4 USD found in a similar urban setting in Chad [43]. Given that 61% of Congolese live of less than US$ 1.9 a day [56], dog vaccination is not affordable by a large part of the population. Willingness to pay surveys always have limitations because actual behavior might differ from hypothetical claims when confronted with real payment and we recommend further studies (e.g., double bounded dichotomous choice contingent valuation [57]) to validate dog owner’s willingness to pay for dog rabies vaccine. Freedom of dog rabies should be declared a public good and dog rabies vaccination subsidized by the government to achieve dog rabies elimination effectively. Development Impact Bond (DIB) financing schemes should be tested for their suitability for dog rabies elimination in the DRC [58]. 

### 4.3. Dog Population Characteristics

Similar to other urban dog populations in sub-Saharan Africa, our dog study population was young (32% puppies, 49% < 1 year) [38,45,47,49,51,54], indicating rapid population turnover, and very few dogs were neutered [38,46,47]. The mean litter size per female dog was 6, higher than the average litter size of 4.7–5.7 reported from other sub-Saharan African countries [37,38,39,59]. Sex ratios skewed towards male dogs are a consistent feature reported in studies on urban dog populations in sub-Sahara Africa [8,38,39,45,46,48,51,54,59,60], with the exception of Lagos State, Nigeria [49]. Possibly, dog owners have a preference for male dogs which are assumed to perform better as guard dogs [61]. Another explanation might be a male-biased birth ratio and a lower female survival rate [62]. We did not observe a predominance of male dogs in our study (sex ratio exactly 1:1), although puppies were not sexed. Hambolu *et al*. [49] explained the predominance of female dogs in Lagos State (Nigeria) as use of dogs for breeding. 

### 4.4. Animal Bite Incidents

Bite victims in this study were exclusively bitten by dogs. Sex and age are two demographic characteristics frequently identified as risk factors for dog bites worldwide, with children and males being disproportionately affected [20,21,63,64,65,66,67,68,69,70,71]. Our finding that 38% of bites occurred in children is in accordance with age distributions of bite victims reported in these studies. We did not record the sex of bite victims. The increased risk of dog bites in children is attributed to the fact that children enjoy playing with dogs but are unable to read their behavioral signals and emotions. Children are at higher risk of rabies because their shorter stature makes them more likely to incur dangerous bites to the neck and face and they may not report bites to caregivers [72]. Therefore, children present an important at-risk population that needs to be educated about the dangers of dog bites and appropriate care measures after a bite. Meta-analyses of educational programs on bite prevention in children found no direct evidence that such programs are effective in reducing bite rates [73,74], but school-based interventions were successful in encouraging children to report bite injuries and increasing knowledge about appropriate PEP [75,76,77]. Almost all bite victims (89%) in this study were bitten by a familiar dog, meaning the majority of biting animals could be identified and subsequently put under observation or tested for rabies to inform decision-making on the need for PEP.

Dog bite incidence can be used to indirectly estimate human rabies deaths [20]. Epidemiological data on dog bites are often available from health facilities but are likely to be biased because not all bite victims seek care after a bite. To date, little data is available on community-level incidence of dog bites. In this study, we found an annual dog bite incidence of 5.2 per 1000 population. When extrapolated to the population of the city of Matadi, we would expect 1500–1600 dog bite victims per year. The dog bite incidence found in this study is considerably higher than the annual dog bite incidence of 0.5/1000 persons and 1.2/1000 persons reported in Mali and Côte d’Ivoire, respectively [78,79], but is similar to the annual dog bite incidence documented in Chad. Higher incidences were reported from rural areas in Asia, for example, 72.9/1000 population in Bangladesh [80], 48.4/1000 population in Cambodia [81], and 17–19.6/1000 population in India [67,82,83]. Previous studies have shown that dog/human ratios are typically higher in rural settings [8] potentially resulting in more frequent dog-human interactions and thus an increased risk of bite incidents. 

### 4.5. Community Knowledge

Only half of our study population had heard of rabies. This proportion is considerably lower than reported in other sub-Saharan African countries, where 76%–99% of respondents had heard of rabies [37,47,84,85,86,87]. The majority of respondents knew that rabies is primarily transmitted through infected dogs (94%) via bites (88%). Rabies seems to be typically associated with dogs, reflected in the local name of the disease “Maladie Ya Mbwa” meaning “disease of the dog”, and is similar to the Arabic term الكَلْبِ داء (“dog disease”) used in Sahel countries. Although respondents were familiar with rabies transmission, they were unaware of disease symptoms, prevention, and control. In order to detect suspected rabid animals within the community and take appropriate measures, such as informing the veterinary service, the public needs to be aware of the signs of rabies. Study participants who claimed to have seen a rabid animal had higher odds of correctly describing the symptoms of rabies in animals indicating that the disease really is prevalent in the study area. However, a minority (1%) mentioned paralysis as a possible symptom of rabies in animals, despite being typical for the paralytic (or dumb) form of rabies. This may result in cases of paralytic rabies going undetected. We did not find an association between knowing a person who had died of rabies and correctly describing the symptoms of human rabies. Clinical manifestation of human rabies is non-specific, which can lead to misdiagnosis of the disease [88]. It is particularly worrisome that 86% of respondents did not know that rabies is fatal after onset of symptoms. About half of respondents knew of dog vaccination and only 3% knew of PEP as a preventive measure. While dog vaccination is crucial to control rabies at its animal source, timely administration of PEP is essential to prevent rabies infection after a bite. 

### 4.6. Community Practices

The general poor rabies knowledge is reflected in the community’s practices towards suspected rabid animals and health seeking behavior. Although two-thirds of respondents would kill a suspected rabid animal, only one in five would report the incident to the veterinary service. This will likely affect rabies surveillance efforts because cases will go undetected. A minority of respondents would wash their wound with water and soap if bitten by a dog. Low awareness about first aid measures following a bite seems to be a common feature across sub-Saharan Africa [47,66,87,89,90,91]. Washing a bite wound with water and soap for 15 minutes is a simple but potentially lifesaving first aid measure that can prevent an infection through mechanical removal or inactivation the RABV [92]. The knowledge gap about PEP as a preventive measure aligns with the alarmingly low proportion of respondents (2%) who would seek PEP after a bite. In contrast to other studies [87,93], our study population would not rely on traditional medicine in case of a bite. Even though most respondents would seek medical care after an animal bite, this does not guarantee receiving adequate treatment. For example, in Ghana, 60% of primary health care providers were not aware of the importance of PEP after a bite and 76% of health care facilities did not have human rabies vaccine in stock [93]. Because a high proportion of potential bite victims would seek medical care, health care provider’s knowledge about rabies and management of bite wound should be assessed. The positive correlation between the knowledge score and practices score indicates translation of knowledge into practice. 

Like most observational studies, this study has potential limitations. Because of lack of alternatives, random geo-coordinates were used as starting points for the household survey. Therefore, households from less densely populated areas might be overrepresented. As mentioned, the study population represents urban communities. Dog/human ratios in rural areas are generally higher, therefore expanding data collection to rural areas is necessary to allow extrapolation to the entire DRC. In addition, we did not investigate the unowned dog population, although their number is considered to be less than 2% of the total dog population [31]. Furthermore, we asked respondents about bite incidents in the last three years, and respondent-reported data could be subject to recall bias. Finally, we did not evaluate respondents’ source of rabies information. This could target common sources, for example, media, health workers, or community leaders, for future awareness raising campaigns and effective information channels. Ideally, the KAP study should be repeated after an awareness raising campaign to assess the impact on the community’s knowledge and monitor changes in the practices. 

## 5. Conclusions

Given the large proportion of unvaccinated, free-roaming dogs coupled with a high dog bite incidence and poor community knowledge about rabies prevention and control, human rabies deaths are likely to occur in this community. Even though findings from this study indicate that the owned dog population is accessible for parenteral vaccination, lack of disease awareness could negatively influence community participation in future dog mass vaccination campaigns. This highlights the urgent need for a public sensitization campaign to promote appropriate prevention (washing bite wounds and PEP) and control (dog vaccination) measures in this community. Our results can support veterinary and public health officials in planning dog vaccination and awareness raising campaigns in the DRC.

## Figures and Tables

**Figure 1 vaccines-07-00098-f001:**
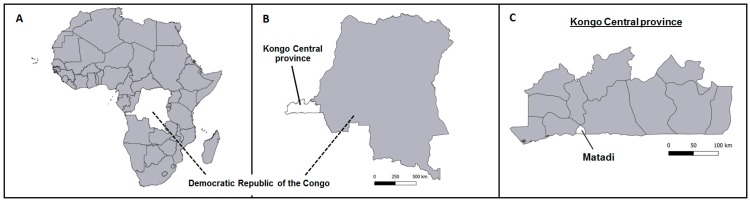
Maps indicating the Democratic Republic of the Congo (**A**) and the Kongo Central province (**B**) with the study site Matadi (**C**).

**Figure 2 vaccines-07-00098-f002:**
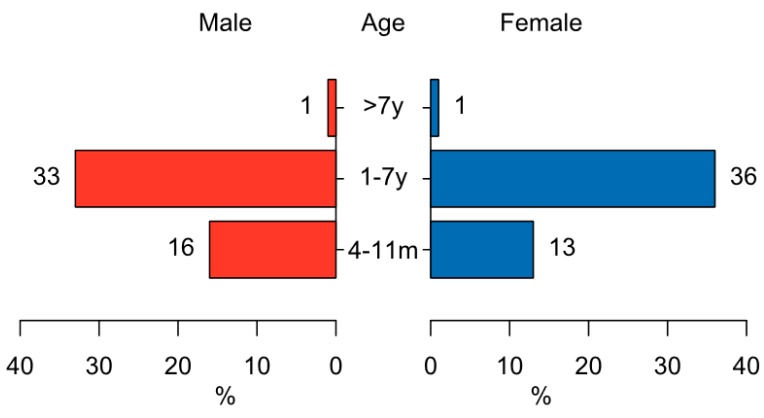
Age and sex distribution of surveyed dogs older than 3 months. Note that age categories are not distributed equally.

**Figure 3 vaccines-07-00098-f003:**
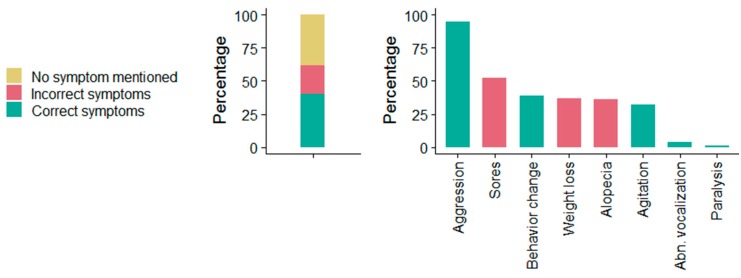
Animal rabies symptoms mentioned by the study participants in Matadi, Democratic Republic of the Congo (DRC) in May 2017. The left plot displays the proportion of respondents that did not mention any rabies symptoms and the proportion of respondents that stated correct and incorrect symptoms. The right plot displays frequently mentioned correct and incorrect animal rabies symptoms.

**Figure 4 vaccines-07-00098-f004:**
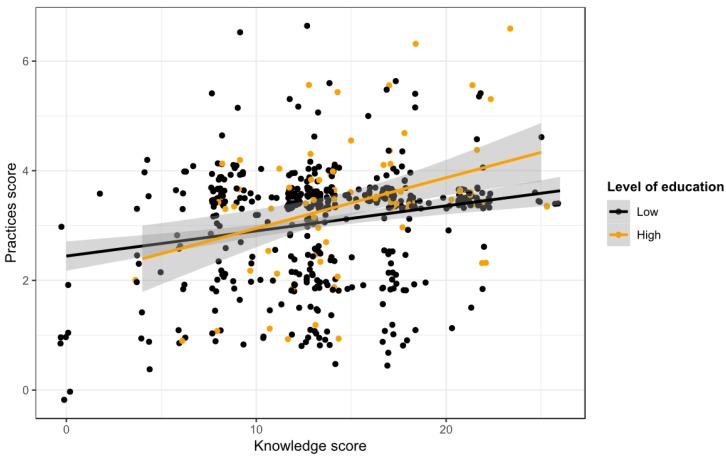
Relationship of knowledge and practices score by level of education. Small random noise was added to avoid over-plotting.

**Table 1 vaccines-07-00098-t001:** Characteristics of the 1056 households and respondents surveyed in Matadi, Democratic Republic of the Congo (DRC) in May 2017.

Variable	Overall (*n* = 1056)
**Sex of respondent**	
Female	669 (63.4%)
Male	387 (36.6%)
**Age of respondent (years)**	
16-29	259 (24.5%)
30-39	281 (26.6%)
>40	516 (48.9%)
**Level of education of respondent**	
None	23 (2.2%)
Primary	85 (8.0%)
Secondary	818 (77.5%)
Tertiary	130 (12.3%)
**Socioeconomic status**	
Middle	920 (87.1%)
Low	136 (12.9%)
**Position of respondent in the household**	
Wife of head of household	464 (43.9%)
Head of household	317 (30.0%)
Child of head of household	211 (20.0%)
Other relative of head of household	64 (6.1%)
**Occupation of respondent**	
Private sector	603 (57.1%)
Public sector	316 (29.9%)
Unemployed	101 (9.6%)
Retired	36 (3.4%)
**Dwelling ownership status**	
Owner	629 (59.6%)
Tenant	427 (40.4%)
**Source of water**	
Improved	893 (84.6%)
Unimproved	163 (15.4%)
**Waste disposal**	
Open	707 (67.0%)
Closed	349 (33.0%)
**Livestock ownership**	
No	769 (72.8%)
Yes	287 (27.2%)
**Residence**	
Peri-urban	557 (52.7%)
Urban	499 (47.3%)
**History of bite incident**	
No	964 (91.3%)
Yes	92 (8.7%)
**Dog ownership**	
No	956 (90.5%)
Yes	100 (9.5%)

* Source of water was categorized as improved or unimproved according to WHO Water, sanitation and hygiene (WASH) standards [41]. # Open waste disposal: public/private waste disposal site, burying waste/ closed waste disposal: incineration of waste.

**Table 2 vaccines-07-00098-t002:** Uni- and multivariable generalized estimating equation models for binomial outcome variables to determine household characteristics associated with dog ownership in Matadi, Democratic Republic of the Congo (DRC) in May 2017. OR: odds ratios; CI: confidence interval.

Variables		% (Npos)	OR	95% CI	*p*-value	Adj OR	95% CI	*p*-value
**Socioeconomic status**	Middle	10% (90/920)	reference					
	Low	7% (10/136)	0.73	0.37–1.44	0.366			
**Dwelling ownership status**	Tenant	5% (21/427)	reference			reference		
	Owner	13% (79/629)	2.78	1.65–4.68	**<0.001**	2.37	1.36–4.15	**0.002**
**Livestock ownership**	No	7% (55/769)	reference			reference		
	Yes	16% (45/287)	2.41	1.53–3.8	**<0.001**	2.01	1.25–3.23	**0.004**
**Residence**	Peri-urban	10% (56/557)	reference					
	Urban	9% (44/499)	0.87	0.54–1.39	0.55			

**Table 3 vaccines-07-00098-t003:** Uni- and multivariable generalized estimating equation models for binomial outcome variables to determine respondent and household characteristics associated with rabies knowledge in Matadi, Democratic Republic of the Congo (DRC) in May 2017.

Variable		% (Npos)	OR	95% CI	*p*-value	Adj OR	95% CI	*p*-value
**Sex**	Female	29% (86/299)	reference			reference		
	Male	41% (103/252)	1.71	1.25–2.34	**0.001**	1.44	1.02-2.03	**0.038**
**Age**	16-29	24% (28/118)	reference			reference		
	30-39	31% (39/128)	1.41	0.78–2.53	0.252	1.63	0.85–3.11	0.14
	>40	40% (122/305)	2.14	1.28–3.58	**0.004**	2.21	1.29–3.78	**0.004**
**Level of education**	Low	32% (149/466)	reference			reference		
	High	47% (40/85)	1.89	1.19–3.01	**0.007**	1.87	1.22–2.87	**0.004**
**Socioeconomic status**	Middle	36% (176/493)	reference					
	Low	22% (13/58)	0.52	0.24–1.15	0.107	0.49	0.22–1.1	0.084
**Livestock ownership**	No	32% (126/392)	reference			reference		
	Yes	40% (63/159)	1.39	0.87–2.21	0.17	1.14	0.71–1.83	0.599
**Residence**	Peri-urban	41% (126/311)	reference			reference		
	Urban	26% (63/240)	0.52	0.29–0.95	**0.033**	0.5	0.28–0.91	**0.024**
**History of bite incident**	No	34% (170/494)	reference			reference		
	Yes	33% (19/57)	0.95	0.53–1.73	0.874			
**Dog ownership**	No	35% (173/491)	reference			reference		
	Yes	27% (16/60)	0.67	0.39–1.14	0.14	0.59	0.34–1.02	0.06

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
