# Peer review of "Dog Ecology, Bite Incidence, and Disease Awareness: A Cross-Sectional Survey among a Rabies-Affected Community in the Democratic Republic of the Congo"

_vaccines, 2019, doi:10.3390/vaccines7030098_

Round 1

Reviewer 1 Report

Dear authors,

Major comments:

This manuscript provides wide range of new and important information associated with dog and human rabies in a selected study area of the Democratic Republic of the Congo. The methodology employs good standard of sampling and analyses, although there are several points to be addressed. Results section needs to be edited and some statistics are better to be added to make use of data. Discussion part consists of several sections and each section has substantial volume. By going through all the parts of discussion, readers can understand the context of important topics for rabies control.

Specific comments:

<2. Methods>

2.3. Sample size calculation and household sampling procedure

Line 124. Random sampling of geo-coordinates: I think the probability of selecting households is reciprocal to the population density. Meaning the samples less represent densely populated areas. Any justification or limitation statement here?

2.5. Ethical approval

Line 192. ‘and ethical clearance’ – ‘and’ is Italic and please correct it.

<3. Result>

3.2. Dog ownership and dog-keeping practices

Line 213-214: Obviously the distribution of number of dogs per household is skewed. SD does not tell readers very well. Could you show median and either 1at and 3rd quartiles, or 2.5% - 97.5%ile, please?

3.3.1. Dog vaccination and accessibility for parenteral vaccination

Line 260-261: It would be helpful if you could provide percentages to before 18/25 and 6/18, respectively.

3.5.1. Factors associated with rabies knowledge

Table 4 - Ideally, more context-based socio-economic factors should be shown. I would suspect that urban areas have tenant households in which the level of education and income are low. So either scrutinizing causal web in modelling, or performing additional analyses to show confounding structure can be recommended.

3.5.4. Relationship between rabies knowledge and practices

Line 331: ANCOVA (practice ~ knowledge + education + knowledge:education) can tell the difference in the slopes for practices score predicted by knowledge. Both parametric and non-parametric methods are available.

<4. Discussion>

4.1. Dog ownership and dog-keeping practices

Dog-human ratio: If I understood correctly that geo-coordinates sampling is assuming complete random distribution of households in the study area, the samples over represent peri-urban areas, as the household density is higher in urban areas. The proportions of households with dogs are not significantly different between ueban and peri-urban, but could you double check that the number of dogs per households are not different between two settings? If they are different, may be better to adjust the effect.

4.2. Dog vaccination and accessibility

It is not written how willingness-to-pay was measured. Simple direct question may not be accurate. At least mentioning several statistical approaches such as double-bounded dichotomous choice is recommended to inform readers about the limitation.

Figure 1: It is better to label three panels (ex. a, b, c), and add explanations. In the second panel, Kongo Central Province should be indicated. Please add distance scale to the country map as well.

Table 2: Is this table necessary? Information was provided in the texts.

Figure 2: It takes time for readers to follow the calculation of months, to check that they are the same age categories in the texts. You should be able to label the age categories like (4-11m, 1-7y, <7y).

Table 4: Please add the information that it is multivariable analysis result, to the title.

Author Response

Major comments:

This manuscript provides wide range of new and important information associated with dog and human rabies in a selected study area of the Democratic Republic of the Congo. The methodology employs good standard of sampling and analyses, although there are several points to be addressed. Results section needs to be edited and some statistics are better to be added to make use of data. Discussion part consists of several sections and each section has substantial volume. By going through all the parts of discussion, readers can understand the context of important topics for rabies control.

We thank the reviewer for this positive overall approval of our work and appreciate the helpful and constructive comments. We have carefully revised the manuscript in light of the reviewer's comments and suggestions. Below, please find our point-by-point response, indicating how and where in the manuscript (line numbers) changes have been made.

Specific comments: 2. Methods>

2.3. Sample size calculation and household sampling procedure

Line 124. Random sampling of geo-coordinates: I think the probability of selecting households is reciprocal to the population density. Meaning the samples less represent densely populated areas. Any justification or limitation statement here?

The reviewer raised an important point. Bias may be introduced because households in sparsely populated areas have a higher probability to be selected compared to households from densely populated areas. However, the impact of this bias is likely limited compared to other potential sources of bias because the population density in Matadi does not vary much in urban centers, e.g. there are no shopping zones with a very low population density. Still, we agree with the reviewer that population density is higher in the city center compared to peri-urban areas. We, therefore, included the following sentence in the discussion:

"Because of lack of alternatives, random geo-coordinates were used as starting points for the household surveys. Therefore, households from less densely populated areas might be overrepresented. " (Line 503-505)    

2.5. Ethical approval

Line 192. ‘and ethical clearance’ – ‘and’ is Italic and please correct it.

We revised as requested. In addition, we added the reference number of the study document submitted to and approved by the ethical committee of Ministry of Public Health at the Clinique Ngaliema. Lines 194-195.

<3. Result>

3.2. Dog ownership and dog-keeping practices

Line 213-214: Obviously the distribution of number of dogs per household is skewed. SD does not tell readers very well. Could you show median and either 1at and 3rd quartiles, or 2.5% - 97.5%ile, please?

We revised as requested and present the median and 1st and 3rd quartile. Line 217-218

3.3.1. Dog vaccination and accessibility for parenteral vaccination

Line 260-261: It would be helpful if you could provide percentages to before 18/25 and 6/18, respectively.

We revised as requested. Line 266-267  

3.5.1. Factors associated with rabies knowledge

Table 4 - Ideally, more context-based socio-economic factors should be shown. I would suspect that urban areas have tenant households in which the level of education and income are low. So either scrutinizing causal web in modelling, or performing additional analyses to show confounding structure can be recommended.

We principally agree with the reviewer but would like to point out that the variable socioeconomic status has been adapted from the DHS methodology together with the local staff. Variables such as income are difficult to capture in this setting. Causal modeling using structural equation models or similar would be beyond the scope of the manuscript.

3.5.4. Relationship between rabies knowledge and practices

Line 331: ANCOVA (practice ~ knowledge + education + knowledge:education) can tell the difference in the slopes for practices score predicted by knowledge. Both parametric and non-parametric methods are available.

We did not present the formal assessment of the interaction of education level and knowledge score because both variables are correlated as shown in table 4. We updated the figure 4. Line 343-344

<4. Discussion>

4.1. Dog ownership and dog-keeping practices

Dog-human ratio: If I understood correctly that geo-coordinates sampling is assuming complete random distribution of households in the study area, the samples over represent peri-urban areas, as the household density is higher in urban areas. The proportions of households with dogs are not significantly different between ueban and peri-urban, but could you double check that the number of dogs per households are not different between two settings? If they are different, may be better to adjust the effect.

The mean number of dogs per household (all households) is 0.19 vs 0.14 in peri-urban vs urban areas. Although, the difference is small one could adjust for it in the estimation. In fact, we presented the data in a previous version but decided to exclude the information because the confidence limits for the subgroups were broad and adjustment for the overall dog human ratio is not possible because we do not have the required denominator data, i.e. it is unknown how many persons live in the urban and the peri-urban areas of the city.

4.2. Dog vaccination and accessibility

It is not written how willingness-to-pay was measured. Simple direct question may not be accurate. At least mentioning several statistical approaches such as double-bounded dichotomous choice is recommended to inform readers about the limitation.

We added the following text to the new manuscript highlighting the limitations of willingness to pay studies: “Willingness to pay surveys always have limitations because actual behavior might differ from hypothetical claims when confronted with real payment and we recommend further studies (e.g. double bounded dichotomous choice contingent valuation [59]) to validate dog owner’s willingness to pay for dog rabies vaccine.” Lines 417-421

Figure 1: It is better to label three panels (ex. a, b, c), and add explanations. In the second panel, Kongo Central Province should be indicated. Please add distance scale to the country map as well.

We revised as requested. Line 110 

Table 2: Is this table necessary? Information was provided in the texts.

We agree with the reviewer that the table provides double information and removed it from the manuscript. Line 222

Figure 2: It takes time for readers to follow the calculation of months, to check that they are the same age categories in the texts. You should be able to label the age categories like (4-11m, 1-7y, <7y).

We thank the reviewer for this remark and updated the figure accordingly. Line 262

Table 4: Please add the information that it is multivariable analysis result, to the title.

We thank the review for this remark. The table shows results from uni- and multivariable analysis but the table was cut off due to portrait formatting of the corresponding page. We changed the formatting of the page (landscape) and title accordingly (Line 321-322). Likewise, we adjusted the title and page layout of table 3 (Line 250-251).

Reviewer 2 Report

Very well written, pleasure to read. Additional citations could be useful in the background section, however besides that stylistically sound besides minor stylistic changes being required e.g. "21,500" instead of "21’500".

Author Response

Very well written, pleasure to read. Additional citations could be useful in the background section, however besides that stylistically sound besides minor stylistic changes being required e.g. "21,500" instead of "21’500".

We thank the reviewer for this positive overall approval of our work. The numbers were revised accordingly throughout the document. Line 34 and Line 108